# Transdermal Flunixin Meglumine as a Pain Relief in Donkeys: A Pharmacokinetics Pilot Study

**DOI:** 10.3390/metabo13070776

**Published:** 2023-06-21

**Authors:** Amy K. McLean, Tara Falt, Essam M. Abdelfattah, Brittany Middlebrooks, Sophie Gretler, Sharon Spier, David Turoff, Francisco Javier Navas Gonzalez, Heather K. Knych

**Affiliations:** 1Department of Animal Science, University of California Davis, Davis, CA 95616, USA; tdfalt@ucdavsi.edu; 2World Donkey Breeds Project, Faculty of Veterinary Sciences, University of Córdoba, 14071 Córdoba, Spain; fjng87@hotmail.com; 3Equitarian Initiative, Stillwater, MN 55028, USA; dturoff@fmvs.biz; 4Department of Animal Hygiene, and Veterinary Management, Faculty of Veterinary Medicine, Benha University, Moshtohor 13736, Egypt; 5Department of Population Health & Reproduction, School of Veterinary Medicine, UC Davis, Davis, CA 95616, USA; 6Department of Clinical Sciences, Colorado State University, Fort Collins, CO 80523, USA; brittany.middlebrooks@colostate.edu; 7K.L. Maddy Equine Analytical Pharmacology Lab, School of Veterinary Medicine, UC Davis, Davis, CA 95616, USA; srgretler@ucdavis.edu (S.G.); hkknych@ucdavis.edu (H.K.K.); 8Department of Medicine and Epidemiology, School of Veterinary Medicine, UC Davis, Davis, CA 95616, USA; sjspier@ucdavis.edu; 9Department of Genetics, Veterinary Sciences, University of Cordoba, 14071 Córdoba, Spain

**Keywords:** donkeys, flunixin, pharmacokinetics, eicosanoids, pharmacology, transdermal flunixin, pain, welfare

## Abstract

Recent approval of transdermal flunixin meglumine (FM) (Banamine^®^) in cattle has opened the door for the drug’s potential application in other species. Transdermal FM could provide a safe and effective form of pain relief in donkeys. In order to evaluate the pharmacokinetics and effects of FM on anti-inflammatory biomarkers in donkeys, a three-way crossover study design was employed. In total, 6 healthy donkeys were administered transdermal (TD) FM at a dosage of 3.3 mg/kg, and oral (PO) and intravenous (IV) doses of 1.1 mg/kg body weight. Blood samples were collected over 96 h to determine the concentration of flunixin, 5OH flunixin, and eicosanoids (TXB2 and PGF2 alpha) using LC-MS/MS. The results indicated that both flunixin and 5OH flunixin were detectable in blood samples collected during TD. The elimination of the drug was slower following the TD route compared to PO and IV. TD administration significantly decreased TXB2 levels in non-stimulated serum from 1 to 96 h post-administration, while IV and PO resulted in TXB2 reduction for 1 to 8 h. A significant reduction in PGF2 alpha was observed in PO and IV 1 h after administration, while TD resulted in a gradual decline from 4 to 72 h. The study concluded that the off-label use of transdermal FM at 3.3 mg/kg could be effective in controlling inflammation in donkeys.

## 1. Introduction

Improvements in donkey welfare have slowly presented themselves in more recent research ranging from establishing qualitative behavior assessments, to advancing operational and handling techniques, as well as optimizing pain management. Donkeys are stoic animals which means they may endure pain without the display of feelings and complaints. Due to this behavior, a donkey may ultimately mask pain with behaviors, such as sham eating or by only showing slight indications of pain. These subtle behaviors may go unnoticed, thus concealing illness/injury. Additionally, there is a lack of analgesic drugs with marketing authorization for administration to donkeys and a lack of information on dosing rates and dosing intervals when analgesia is provided. Donkeys differ from horses behaviorally, physiologically, and pharmacologically [1]. Therefore, extrapolation of horse dose rates and dosing intervals for use in the donkey may be inappropriate for donkeys and mules. Non-steroidal anti-inflammatory drugs (NSAIDs) are the most commonly prescribed drugs for equine pain management. These medications have anti-inflammatory and analgesic (pain-relieving) properties and work by blocking cyclooxygenase (COX) enzymes (COX-1 and COX-2). NSAIDs block lipoxygenase, thereby inhibiting the synthesis and release of prostaglandins (PGs) and thromboxane that perpetuate inflammation, pyrexia, and pain [2]. Flunixin meglumine (FM) is a NSAID used to treat pain, inflammation, and fever in food producing animals including cattle, swine, and horses [3]. Flunixin has been used in equine clinical practices for a variety of inflammatory and painful conditions, including colic, colitis, endotoxic shock, respiratory disease, ocular disease, general surgery, and laminitis [4]. Flunixin demonstrates superior efficacy compared to phenylbutazone in preventing the clinical manifestations of endotoxemia. However, its effectiveness appears to be comparable to that of ketoprofen [5]. The previous study by Cheng et al. [6] demonstrated that IV FM at a dose rate of 1.1 mg/kg produces potent anti-inflammatory effects in donkeys by inhibition of cyclooxygenase (COX-1 and COX-2). FM is the first and only FDA-approved product for pain control in a food-producing animal [7]. Banamine^®^ (active ingredient, flunixin meglumine) is currently available in injectable (Intravenous = IV), oral (Per os = PO), and transdermal (TD) forms; however, there are varying levels of approval depending both on species as well as country of use. In the US, Banamine^®^ is approved for IV administration in cattle, intramuscular administration in swine, and both injectable and oral use in horses, only because the latter is not considered a food animal in this country. The relatively new TD form that has proven effective in cattle may provide an alternative to administering flunixin meglumine and providing comfort for donkeys. Currently, only oral and injectable forms of flunixin meglumine are approved for use in equids, and their effectiveness is well noted in the literature [6,7,8]. In addition to their stoic demeanors, donkeys can prove to be more challenging to administer medication through intravenous routes due to a thicker muscle layer found covering the jugular vein (the cutaneous coli muscle). Furthermore, they are resistant to poor tasting oral medication and may even hide or not swallow medicine altogether [9,10]. There have been reports of localized swelling, stiffness, and bacterial myositis [11,12] following injectable administration of flunixin meglumine. The transdermal application of this analgesic would be much easier to apply which would revolutionize pain treatment in this overlooked species. Additionally, TD administration reduces stress for donkeys by eliminating the need for handling and restraining during oral and intravenous medications. The previous studies conducted on horses [13], cattle [14], goat [15], and swine [16] indicated that TD FM is rapidly absorbed and has longer half-life compared to IV administration and resulted in prolonged reduction in PGE2 levels and consequently reduced inflammation. The ex vivo models have been utilized to illustrate the impact of NSAIDs on the eicosanoid biosynthesis pathway [8]. In this experimental model, blood samples are obtained at different time intervals following drug administration. These samples are then incubated with calcium ionophore (CI) and lipopolysaccharide (LPS) to induce the activity of enzymes involved in the arachidonic acid cascade. The concentrations of various eicosanoids are subsequently measured and compared to the pre-drug administration sample.

To the best of our knowledge, there are no published reports describing the use of TD FM in donkeys and the comparison of this route of administration with the commonly used PO and IV routes. Therefore, the objectives of our study were to first describe the blood concentrations of flunixin meglumine and its metabolites following TD, IV, and PO administration to donkeys, and secondly to describe the effects of flunixin meglumine on anti-inflammatory biomarkers following administration of three formulations to donkeys (*Equus asinus*) over 96 h time period using an ex vivo model of inflammation. We hypothesized that TD FM would exhibit comparable pharmacokinetics and effects on inflammation biomarkers in donkeys to IV and PO.

## 2. Materials and Methods

### 2.1. Animals and Housing 

A total of 6 healthy standard donkeys, 5 jennies and 1 gelded jack, ranging in age from 4 to 12 years, and weighing between 136 and 158 kg (300 to 350 pounds) were used in the study. All donkeys were obtained from Peaceful Valley Donkey Rescue (Acton, CA, USA) and kept in Russell Ranch (Davis, CA, USA). This study was approved by the University of California, Davis Institutional Animal Care and Use Committee (IACUC) (protocol #22018). All donkeys were determined to be healthy according to a complete blood chemistry panel, a serum biochemistry panel, and a clinical exam.

### 2.2. Study Design

All donkeys received all treatments in the following order using a three-way crossover study design: oral (PO) by dosing syringe in oral cavity (BANAMINE^®^ (flunixin meglumine paste), IV via the jugular vein (Banamine^®^-Injectable Solution, Merck Animal Health, Madison, NJ, USA), and TD along dorsal midline following the label instructions provided by the commercial BANAMINE^®^ TRANSDERMAL product [17]. A minimum two-week washout period was allowed between treatments. Oral and injectable (IV) flunixin was administered at a dose of 1.1 mg/kg of body weight to each donkey averaging 377 mg for PO. Transdermal was administered according to label instructions for cattle, 3.3 mg/kg BW (3 mL/100 pounds) dose for all donkeys. Donkeys received a single transdermal administration of 12 mL of flunixin. The product was applied along midline of the donkey’s back starting behind the withers and ending over the tail head without shaving coat hair. Throughout the course of each trial, all the donkeys were observed for any skin abnormalities and adverse behavioral effects.

### 2.3. Blood Collection and Analysis

Catheters were placed in the jugular vein for the initial 48 h sampling period and upon removal, the remaining samples were collected via direct venipuncture. Whole blood samples were collected 0 (immediately prior to drug administration), 5, 10, 15, 30, 45 min, and 1, 1.5, 2, 3, 4, 5, 6, 8, 12, 18, 24, 30, 36, 48, 60, 72, 96 h post administration of the drug into tubes devoid of anti-coagulant kept at room temperature. The following samples were collected at each timepoint: 1 serum sample (saved in a tube with no anticoagulant or preservative, red top tube) for determination of thromboxane B2 (TXB2) and prostaglandin F2 alpha (PGF2 alpha) concentrations. For determination of TXB2 concentrations, the samples were incubated at 37 °C for 1 h, and subsequently centrifuged, serum harvested, and stored at −20 °C until analyzed by liquid chromatography-tandem mass spectrometry (LC-MS/MS). Two whole blood samples were collected into a tube containing sodium heparin (green top tube) for stimulation with calcium ionophore (CI), lipopolysaccharide (LPS), or methanol. The samples of whole blood were collected and separated into 3 aliquots (5 mL each). A total of 1 aliquot of whole blood was stimulated with CI (dissolved in 2% methanol, final concentration in samples of 10 µM), incubated for 2 h at 37 °C, and plasma harvested by centrifugation. The second aliquot was stimulated with LPS (dissolved in water, final concentration in samples of 50 µg/mL), incubated for 24 h at 37 °C, and plasma harvested by centrifugation. The third aliquot was stimulated with methanol (ME; 2% as vehicle), incubated for 2 h at 37 °C, and plasma harvested by centrifugation. 

In addition, blood samples were taken for eicosanoid analysis at 0 (immediately prior to drug administration), 1, 2, 4, 6, 8, 24, 48, 72, 96 h post administration of the drug. Eicosanoid samples were kept on ice until they were transferred to a laboratory. Blood samples were centrifuged and spun for 15 min × 3000× *g*. Serum was transferred into cryovials and stored at −20 °C until analysis. Concentrations of flunixin and 5-Hydroxy (5OH) flunixin were determined using liquid chromatography-tandem mass spectrometry (LC-MS/MS) and a previously validated method [18]. 

Non-compartmental analysis was used for determination of pharmacokinetic parameters using commercially available software (Phoenix WinNonlin v8.1, Certara, Princeton, NJ, USA). The maximum concentration (C_max_) and time to reach maximum concentration (t_max_) were based on visual inspection of the concentration–time data. The slope of the terminal portion of the curve, lambda *z* (λ*_z_*), was used to calculate half-life (t1/2 λ_z_) using the equation 0.693/λ*_z_*. The area under plasma concentration–time curve/dose (AUC/Dose) was calculated using the linear up–log–down trapezoidal rule.

### 2.4. Statistical Analysis

#### 2.4.1. Flunixin and 5OH Flunixin Concentration Best Fitting Model Determination across Administration Routes

Data were graphically represented as a function of time and depicted using the Graphs add-on in Microsoft Office LTSC Professional Plus 2021 Excel. Linear, Second, Third, Fourth, Fifth, and Sixth Order Polynomial models were tested in order to choose which model reported better fitting properties for flunixin and 5OF flunixin as a function of time. The trendline add-on Microsoft Office LTSC Professional Plus 2021 Excel was used to determine the best fitting models across administration routes (oral, intravenous, or transdermal). Sextic equations or polynomial equations of degree six were found to be the best fitting models when determination coefficient magnitude (R Squared) was considered. All models ascribed to the following general model equation:(1)y=ax6+bx5+cx4+dx3+ex2+fx+g

a * Administration Route Concentration** 6 + b * Administration Route Concentration ** 5 + c * Administration Route Concentration ** 4 + d * Administration Route Concentration ** 3 + e * Administration Route Concentration ** 2 + f * Administration Route Concentration + g.

With a, b, c, d, e, f, and g being the model reference curve shape parameters.

Reference curve shape parameters across administration routes were calculated using the trendline option add-on in Microsoft Office LTSC Professional Plus 2021 Excel. These model reference curve shape parameters were used as the starting values for curve shape parameters iteration for each particular donkey participating in the study and each route through which flunixin and 5OH flunixin were administered.

#### 2.4.2. Flunixin and 5OH Flunixin Six Order Legendre Polynomial Curve Shape Parameters

Individual specific curve shape parameters, dispersion statistics, confidence intervals and determination coefficients for each administration route were calculated after fitting the following software automatized model across administration routes for either flunixin or 5OH flunixin using the nonlinear routine of the regression package in SPSS Statistics, Version 26.0, (IBM Corporation, Armonk, NY, USA). Individual values were obtained using the Split File routine of the Data package of SPSS Statistics, Version 26.0, (IBM Corporation, Armonk, NY, USA).

#### 2.4.3. Parametric Assumptions Testing and Approach Decision

Parametric assumptions were tested to decide on the most appropriate statistical approach to follow to analyze the present data. The Shapiro–Francia W’ test (for 50 < n < 2500 samples), Shapiro–Wilk test (for n < 50 samples), and Levene’s test were used to discard gross violations of parametric assumptions (normality and homoscedasticity). The Shapiro–Francia W’ test was performed using the Shapiro–Francia normality routine of the test and distribution graphics package of the StataCorp Stata MP 16.0 software (StataCorp, College Station, TX, USA). All assumption testing statistical tests, including all Bayesian procedures, were performed using the explore procedure of the descriptive statistics package in SPSS Statistics (Version 26.0, IBM Corp., Armonk, NY, USA).

#### 2.4.4. Bayesian Paired t-Test to Detect Differences in the Mean across Administration Routes

The paired samples *t*-test routine of the Bayesian Statistics package within the Analyze section in SPSS Statistics (Version 26.0, IBM Corp., Armonk, NY, USA) was run, defining administration route as the grouping variable. Flunixin and 5OH flunixin concentrations characterization of posterior distribution and Bayes factor (BF) were estimated. The list of criteria proposed by Depaoli and Van de Schoot [19] were used to detect: (a) possible prior-to-test incidences, (b) post testing but prior to interpreting incidences, (c) to understand the influence of the priors, and (d) to determine the actions to be taken after interpreting the results. Effects of the factors considered are quantified by evaluating their confidence interval in the posterior distribution statistics. Third, the 95% credibility interval shows that there is a 95% probability that these regression coefficients (posterior distribution true value for each covariate and factor) in the population lie within the corresponding intervals. When 0 is not contained in the credibility interval, the effect is evidently not 0 and a significant effect for such s factor is detected. The BF was calculated to determine the probability of both the null hypothesis and the alternative or one model versus the other based on the a priori distribution of the data. This factor quantifies the change in probability from the a priori distribution to the posterior or a posteriori distribution due to the data. The BF was then calculated to determine the validity of the model containing the significant factors and covariates compared to a model that only considers the intercept. The BF is a measure of the strength of evidence and is used instead of *p*-values (frequency approximations) to draw conclusions. The difference with the classical *p*-value is that the BF gives an indication of support for both hypotheses and compares this evidence. Additionally, it says something about the strength of the evidence for the null hypothesis in comparison with the alternative hypothesis in this situation. On the other hand, the *p*-value is only able to support the decision of rejecting the null hypothesis or not, according to Schalken et al. [20]. A large BF implies that the evidence favors the alternative hypothesis compared to the null hypothesis. The levels commonly used to define evidence significance levels are established following the premises of Jeffreys [21] and Lee, and Wagenmakers [22]. 

#### 2.4.5. Effect of the Flunixin Administration Route on Eicosanoids

Statistical analyses of selected eicosanoids, including TXB2, and PGF2 alpha were performed using Stata software to assess the effect of route of administration of flunixin, time point prior and following administration, and interactions between the route of administration and time points. The data of TXB2 and PGF2 alpha were first checked for normality using histogram frequency graphs. The data of TXB2 and PGF2 alpha were not normally distributed, and log transformed using the natural logarithm (ln) function in Stata 15 software (Stata Corp., College Station, TX, USA). Then, data were analyzed using mixed-effects analysis of variance, with the animal as a random effect and route of administration, time points, and their interactions as a fixed effect. After fitting the model, the pwcompare command was used to perform pairwise comparisons of estimated marginal means for fixed effects. A *p*-value of 0.05 or less was considered significant differences. 

## 3. Results

### 3.1. Bayesian Paired *t*-Test to Detect Differences in the Mean across Administration Routes

Serum concentrations of flunixin were higher with both IV and oral routes during the initial 8 h (Figure 1 and Figure 2; Appendix A). Oral flunixin had the highest concentrations (ng/mL) between 30 min and 5 h, while injectable flunixin peaked from 0.08 min to 3 h after administration. Comparing the concentrations of different forms of flunixin, both IV and oral routes had significantly (*p* < 0.01) higher concentrations from 0.08 min to 18 h than transdermal administration (see Figure 1 and Figure 2). IV concentrations were higher than both oral and transdermal from hours 18 to 60 (Figure 1 and Figure 2). Transdermal flunixin had the highest concentrations between 4 and 6 h but was measurable as early as 15 min and as long as 96 h after administration when compared to oral and injectable concentrations (Figure 1 and Figure 2). 

When comparing metabolite concentrations of flunixin (5-Hydroxy Flunixin), IV 5OH flunixin first appeared at 0.08 (min) hours and remained detectable through 72 h, with the highest peak from 0.08 (min) (to 1 h (Figure 3 and Figure 4). The concentrations of 5OH flunixin following oral flunixin administration were found at higher concentrations from 0.08 (min) hours to 12 h. Transdermal 5OH flunixin concentrations were detectable at 0.5 h until 96 h, with peak concentration between 4 and 24 h and highest concentration at 6 h (Figure 3 and Figure 4). 

### 3.2. Different Pharmacokinetic Parameters following Oral, Intravenous, and Transdermal Administration of Flunixin Meglumine

Regarding the pharmacokinetic values across the three forms of flunixin, the maximum concentration (C_max_, mean ± SE) of 9839.42 ± 452.12 ng/mL was found for IV flunixin, while the mean C_max_ for oral, and transdermal administration of flunixin was found to be 1976.8 ± 463.63 and 161.46 ± 33.89 ng/mL, respectively (Table 1). The mean doses for flunixin varied from 2.75 mg oral to 4.38 mg transdermal (due to transdermal label instructions suggesting a higher dose) (see Table 1). When measuring concentrations in the blood the maximum amount of time to detect flunixin was found to be fastest in intravenous form at 0.08 h and longest with transdermal 5.83 h. Half-life was reported to be shortest in transdermal (0.04 1/h) compared to the other 2 forms and longest in IV (0.13 1/h) (see Table 1). In addition, significant differences were found in half-life, time, and maximum concentration when comparing pharmacokinetic parameters for oral and IV versus transdermal administration of flunixin in donkeys with (*p* < 0.04) (Appendix A). 

### 3.3. Effect of the Flunixin Administration Route on Eicosanoids

The ex vivo models used in this study demonstrated the anti-inflammatory effects of flunixin meglumine following different routes of administration specifically how flunixin affected the eicosanoid biosynthesis pathway (TXB2, and PGF2 alpha). Our results showed that the concentration of TXB2 in non-stimulated serum samples decreased significantly (*p* < 0.001) after the transdermal administration of flunixin meglumine, starting from 1 h post-administration and continuing until 96 h, compared to the baseline (Figure 5A,B). Following IV and oral administration of flunixin meglumine, there was a significant reduction in TXB2 concentrations relative to baseline at 1 h post-drug administration and lasting for up to 8 h (Figure 5A,B).

Following stimulation with LPS, Thromboxane B2 was significantly reduced in IV and oral groups, relative to baseline, starting at 1 h post-administration and until 8 h (*p* < 0.001). While in transdermal groups, the TXB2 significantly reduced, relative to baseline, at 4 h post-administration and continued low until 72 h (*p* < 0.001; Figure 6A,B).

Our results showed that following IV and oral administration of flunixin meglumine, there was a significant reduction in PGF2 alpha concentrations relative to baseline. This reduction began at 1 h post-drug administration and lasted for up to 8 h (Figure 2). In contrast, transdermal administration of flunixin meglumine resulted in a gradual decline in PGF2 alpha concentrations started at 4 h post-administration and lasting for up to 72 h post-drug administration. The inhibition of PGF2 alpha was shorter-lived following IV and oral administration as compared to transdermal administration (Figure 7). Regarding the safety of TD FM administration, we solely monitored the donkeys for any signs of skin redness or hair loss at the application site. However, no adverse dermal effects, such as hair loss or skin redness, were observed in any of the enrolled donkeys following TD administration.

## 4. Discussion

Donkeys are the forgotten equid of research and differ significantly from the horses that their treatments are modeled after. They are traditionally stoic animals that are well known for masking discomfort and pain or presenting it with subtle or less recognizable behavioral indicators [23]. In many cases, injuries or illnesses in donkeys often go unnoticed until they have reached advanced stages [24]. When a donkey is recognized to be suffering, the subsequent challenge is to administer a pain medication that is both effective and suitable for their needs. Several studies have evaluated the effectiveness of NSAIDs such as flunixin meglumine in equids, but only in their injectable or oral forms [25,26,27]. One study suggested that owners may misinterpret pain-related behaviors in donkeys and coupled with a lack of information on dosing rates and intervals in donkeys, may increase the challenge of effectively using non-steroidal anti-inflammatories drugs such as flunixin in donkeys [6]. Therefore, the main objective of this study was describing the pharmacokinetics of FM in donkeys following trans-dermal administration of an FDA-approved product for cattle. In our study, the maximum blood concentration of FM following TD administration in donkeys was an average of 161.46 ± 33.89 ng/mL, which was achieved within 5.83 h. Previous reports describing blood concentrations of FM following transdermal administration are highly variable between species. The maximum blood concentrations of FM following TD administration were reported as 515.6 (range: 369.7–714.0) ng/mL, 1170 ng/mL, 14.61 ± 7.85 ng/mL, and 134 ± 42.0 ng/mL in Thoroughbred horses, Holstein calves, sows, and goats, respectively [13,14,15,28]. In donkeys, the C_max_ was lower than that reported for cattle and horses following administration of the same dose of TD FM. Interestingly, T_max_ was much faster in donkeys (5.83 h) compared with that reported for horses (8–12 h), but slower than cattle (2.14 h). 

To evaluate the anti-inflammatory properties of flunixin meglumine after administration, a previously documented ex vivo inflammation model was employed in horses [18,19]. This model involved collecting blood samples from horses before and after the administration of flunixin, which were then stimulated with LPS and CI. Similar to what is observed with inflammation, the addition of LPS leads to an increase in COX-2 which increases PGE2 synthase, and ultimately, increasing concentrations of PGE2 [12]. In LPS-stimulated whole blood, Prostaglandin E2 and PGF2 alpha serve as substitutes for COX-2 activity. Similar to our results, Knych et al. (2020) [8] and Knych et al. (2021) [13] reported that transdermal administration of flunixin meglumine led to significant suppression of PGE2 concentrations, relative to baseline, starting at 4 h and lasting for up to 72 h post-drug administration. 

In agreement with Cheng et al. [6], following IV injection of FM at 1.1 mg/kg in donkeys, the concentration of TXB2 in serum was significantly inhibited from 1 h to 8 h but returned to normal baseline after 24 h from administration. The concentration of TXB2 in non-stimulated serum samples decreased significantly after the transdermal administration of FM at 3.3 mg/kg, starting from 1 h post-administration and continuing until 96 h, compared to the baseline. These findings are consistent with the results obtained by Knych et al. [13], who observed a significant reduction in TXB2 concentrations post-flunixin meglumine administration starting at 1 h and continuing until 24 h, compared to the baseline. Suppression of TXB2 concentrations in non-stimulated and LPS stimulated whole blood is indicative of inhibition of COX-1 and COX-2 enzymes, respectively. These findings suggest that transdermal flunixin meglumine has anti-inflammatory effects in donkeys. Furthermore, the results demonstrate that the suppression of TXB2 blood concentrations lasts longer for up to 72 h post-drug following transdermal administration compared to IV and oral administrations. 

Our study showed reduction in inflammation biomarker PGF2 alpha post TD FM administration started at 1 h and continued until 72 h, compared to the baseline which was longer than other routes. Prostaglandin PGF2 alpha acts as substitutes for COX-2 activity in LPS-stimulated whole blood. The introduction of LPS, which mimics the inflammatory response, results in elevated COX-2 levels, leading to an increase in PGE2 synthase, and ultimately, higher concentrations of prostaglandin. In line with our findings, Knych et al., [8] found that concentrations of PGF2 alpha were significantly reduced until 72 h post-administration of TD FM in horses. The significant reduction of eicosanoids (TXB2 and PGF2 alpha), which are fundamental to the acute inflammation, illustrate that FM is a potentially useful NSAIDs in donkeys. This agrees with results obtained in horses and other species [6,8,15,29,30].

Banamine^®^ Transdermal has been proven effective in cattle and has the approval of Food and Drug Administration in the US [25]. This study was the first of its kind to test the efficacy of this product in the donkey. Our results show that there was successful uptake of flunixin through the skin and into the systemic circulation with sustained levels. Results of this investigation are supportive of additional studies to assess the clinical effectiveness of Banamine^®^ Transdermal for the treatment of pain in equids [31,32].

To summarize, our study evaluated the pharmacokinetics, serum concentrations, and anti-inflammatory effects of flunixin meglumine following transdermal, IV, and oral administration in donkeys. While serum concentrations following transdermal administration were lower than those observed with IV and oral administration, transdermal administration appears to produce sufficient concentrations to induce anti-inflammatory effects for 24–72 h. Based upon this study and previous studies in donkeys, suggested therapeutic doses for flunixin are 1.1mg/kg for oral and IV, and 3.3 mg/kg for off-label use of transdermal. Additional studies are needed to continue to establish the therapeutic concentrations of transdermal Banamine^®^ in donkeys. In order to mimic the field condition, we administered the TD FM directly on the hair coat over the topline of the donkeys without shaving. We observed no adverse dermal effects, such as hair loss or skin redness, in the donkeys treated with topical medication. However, it is important to conduct additional tolerance and toxicological studies before recommending appropriate dosage rates.

## 5. Study Limitations

Regarding the safety of TD FM for donkeys, we did not measure outcomes to assess the toxicity of FM in donkeys. Considering that TD FM is systemically absorbed, it implies that donkeys may be susceptible to the common side effects associated with NSAIDs administered through other routes, including gastric ulceration, dorsal colitis, and disruption of the intestinal microbiome. Therefore, further studies are required to provide crucial insights into the safety and efficacy of TD FM in donkeys, enabling better understanding and appropriate management of potential adverse effects. In our study, we only tested one transdermal dose of FM (3.3 mg/kg). Therefore, it will be beneficial for future studies to test different doses of TD FM to understand more about its clinical significance and relevance in donkeys.

## 6. Conclusions

Our study showed that in comparison to other routes of administration, such as oral or intravenous, the off-label use of transdermal flunixin meglumine in donkeys resulted in sufficient concentrations of serum flunixin, 5OH flunixin, and eicosanoids to induce anti-inflammatory effects for a duration of 24–72 h. Transdermal Banamine^®^ has the potential to be a useful non-steroidal anti-inflammatory drug that can improve the welfare of millions of donkeys worldwide by being an easier and safer means of administration. Further research is needed to provide more information on the implications of transdermal Banamine^®^ in donkey welfare and pain control.

## Figures and Tables

**Figure 1 metabolites-13-00776-f001:**
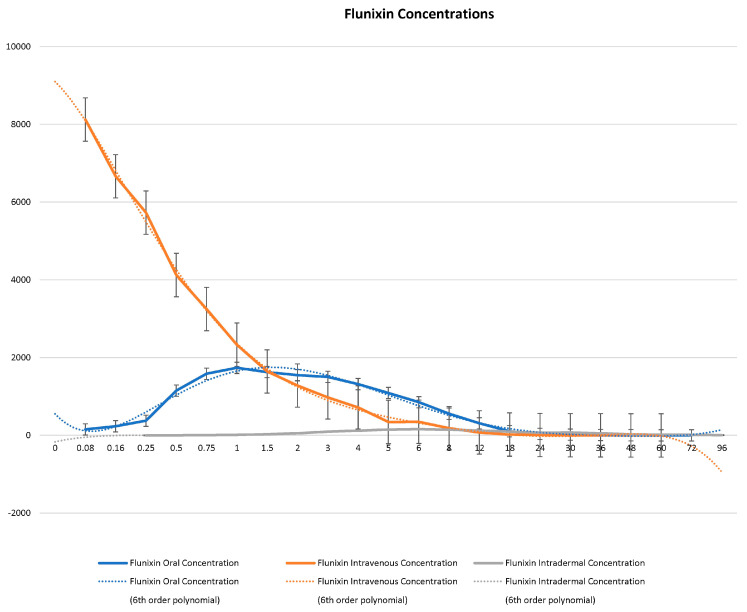
Graphic representation for oral, intravenous, and transdermal concentrations of flunixin ng/mL in donkeys and their corresponding 6th order polynomial trendlines over 96 h post administration.

**Figure 2 metabolites-13-00776-f002:**
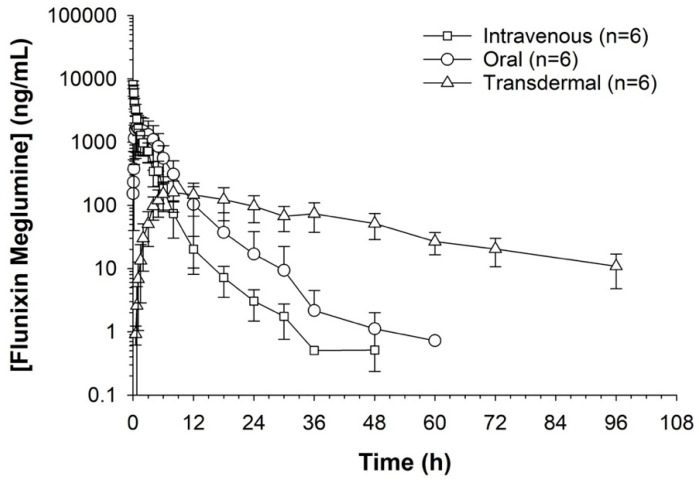
Measuring flunixin (ng/mL) in blood samples from donkeys after flunixin was administered in three different methods: oral through oral cavity, injectable through catheter through intravenous injection, and transdermal by application directly to the dorsal midline.

**Figure 3 metabolites-13-00776-f003:**
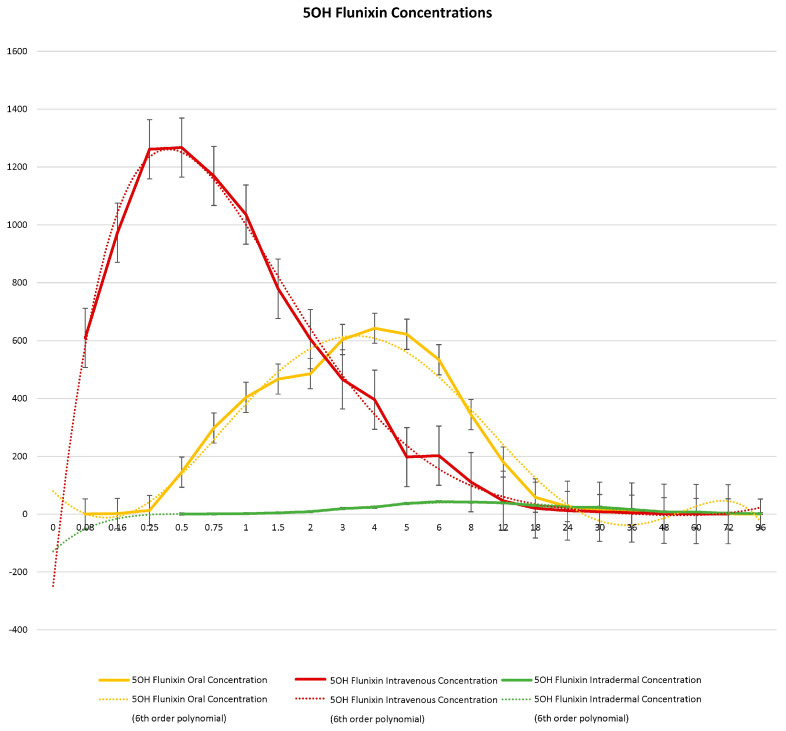
Graphic representation for oral, intravenous, and transdermal concentrations of flunixin metabolite 5OH flunixin (ng/mL) in donkeys and their corresponding 6th order polynomial trendlines over 96 h post administration of flunixin.

**Figure 4 metabolites-13-00776-f004:**
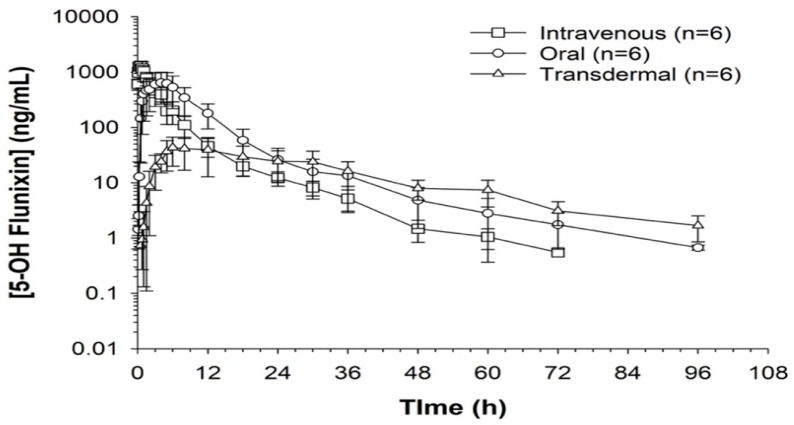
Flunixin administered to donkeys in three routes: oral through oral cavity, injectable through catheter in jugular vein, and transdermal by application directly to the dorsal midline when measuring concentrations of metabolite 5-OH flunixin (ng/mL) in blood samples over a 96 h period.

**Figure 5 metabolites-13-00776-f005:**
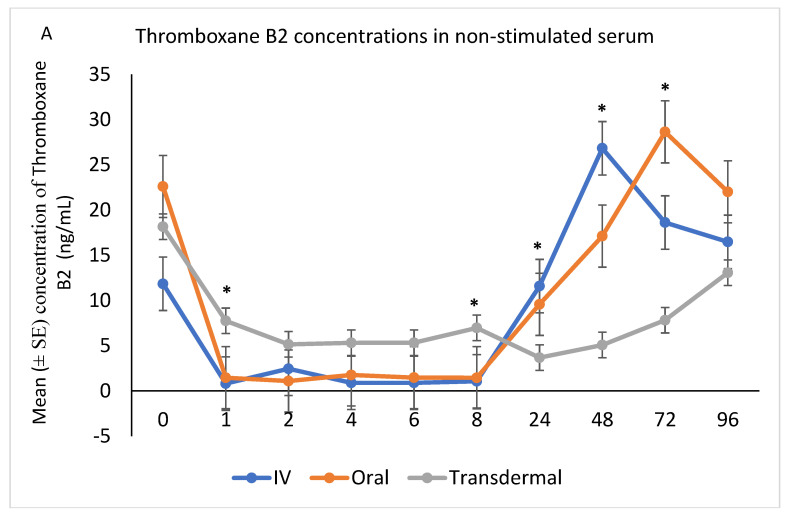
(**A**,**B**): Effects of the route of administration of flunixin (Intravenous = IV; Oral; transdermal) on Thromboxane B2 (TXB2) in non-stimulated blood samples following administration to 6 healthy standard donkeys. Whole blood samples were collected 0 (immediately prior to drug administration = baseline), 1, 2, 4, 6, 8, 24, 48, 72, 96 h post administration of the drug. * Mean values in the same time points differ significantly (*p* < 0.05).

**Figure 6 metabolites-13-00776-f006:**
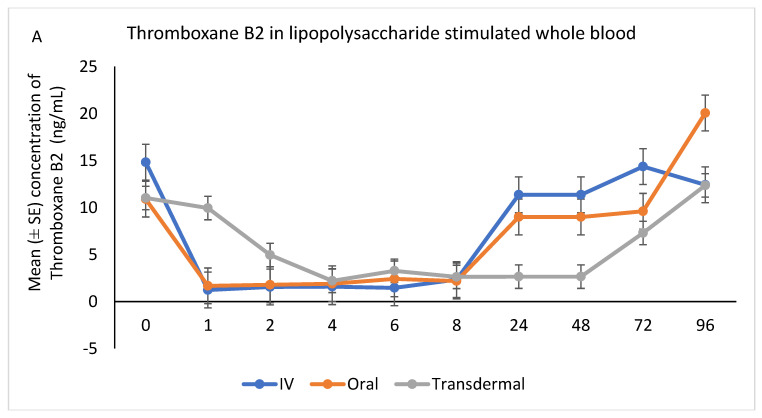
(**A**,**B**): Effects of the route of administration of flunixin (Intravenous = IV; Oral; transdermal) on Thromboxane B2 (TXB2) in lipopolysaccharide stimulated whole blood following administration to 6 healthy standard donkeys. Whole blood samples were collected 0 (immediately prior to drug administration), 1, 2, 4, 6, 8, 24, 48, 72, 96 h post administration of the drug. * Mean values in the same time points differ significantly (*p* < 0.05).

**Figure 7 metabolites-13-00776-f007:**
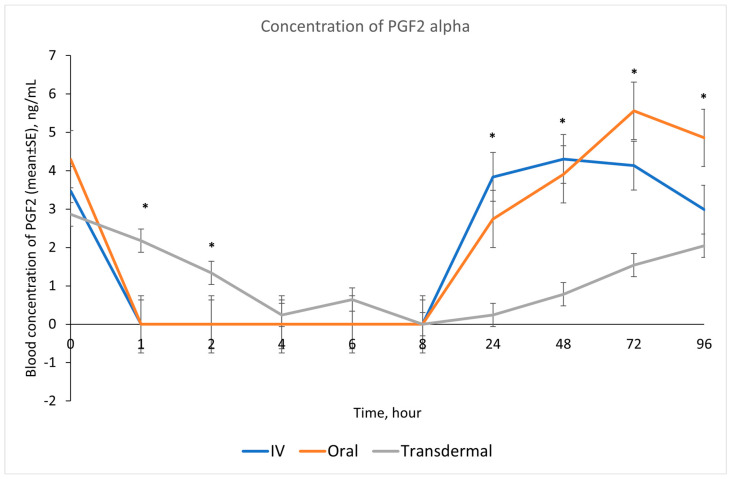
Effects of the route of administration of flunixin (Intravenous = IV; Oral; transdermal) on the means (± SE) of Prostaglandin F2 alpha (PGF2 alpha) concentrations in lipopolysaccharide stimulated whole blood following administration to 6 healthy standard donkeys. Whole blood samples were collected 0 (immediately prior to drug administration), 1, 2, 4, 6, 8, 24, 48, 72, 96 h post administration of the drug. * Mean values in the same time points differ significantly (*p* < 0.05).

**Table 1 metabolites-13-00776-t001:** The descriptives of the pharmacokinetic parameters including the area under plasma concentration–time curve/dose (AUC/Dose), concentration maximum (C_max_ (ng/mL), time to reach maximum concentration (t_max_ reported in hours), and half-life (Lambda Z, 1/h) of flunixin following intravenous (IV), oral (PO), and transdermal (TD) administration of flunixin meglumine (Banamine^®^) in 6 donkeys.

Route of Flunixin Administration	Values ^1^	AUC/Dose	C_max_	t_max_	Drug Half-Life (t1/2)	AUC_INF_	AUC_last_
	Units	mg	ng/mL	h	1/h	h × ng/mL	h × ng/mL
**IV**	Mean ± SE	1.23 ± 0.02 ^c^	9839.42 ± 452.12 ^a^	0.08 ± 0.00 ^c^	0.13 ± 0.004 ^a^	10,641.68 ± 1096.733 ^a^	10,632.23 ± 1094.69 ^a^
	Median	1.2	9712.673	0.08	0.137	9845.625	9838.74
	95% Conf. Interval	1.18–1.27	8885.51–10,793.33		0.126– 0.145	8327.778–12,955.58	8322.62–12,941.83
**PO**	Mean ± SE	2.75 ± 0.05 ^b^	1976.8 ± 463.63 ^b^	1.33 ± 0.35 ^b^	0.13 ± 0.013 ^a^	12,589.17 ± 3145.094 ^a^	12,583.5–3144.97 ^a^
	Median	2.7	2146.45	1	0.136	11,817.5	11,812.5
	95% Conf. Interval	2.63–2.86	998.61–2954.98	0.59–2.07	0.106–0.164	5953.599–19,224.73	5948.18–19,218.82
**TD**	Mean ± SE	4.38 ± 0.01 ^a^	161.46 ± 33.89 ^c^	5.83 ± 0.16 ^a^	0.04 ± 0.002 ^b^	4521 ± 787.34 ^b^	4383.66–757.05 ^b^
	Median	4.4	171.30	6	0.039	5194	5043
	95% Conf. Interval	4.34–4.42	89.96–232.97	5.48–6.18	0.036–0.045	2859.843–6182.157	2786.42–5980.91
	*p*-value	<0.001	<0.001	0.001	<0.001	0.014	0.013

^1^ AUC/Dose: the area under the plasma drug concentration–time curve (AUC) reflects the actual body exposure to drug after administration of a dose of the drug and is expressed in mg × h/L; C_max_: is the maximum concentration of a drug in the blood after a dose is given; t1/2: is the length of time required for the concentration of a particular substance (typically a drug) to decrease to half of its starting dose in the body; AUCinf: area under the plasma concentration–time curve from time 0 to infinity. AUC was determined from the plasma concentration-time profile using non-compartmental analysis; AUClast: AUC to the last measurable voncentration. ^a–c^ Mean values in the same column with different superscripts differ significantly (*p* < 0.05).

## Data Availability

The data is not publicly available due to privacy restrictions but are available upon request from corresponding authors.

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
