# Peer review of "Transdermal Flunixin Meglumine as a Pain Relief in Donkeys: A Pharmacokinetics Pilot Study"

_metabolites, 2023, doi:10.3390/metabo13070776_

Round 1
Reviewer 1 Report
The article is well written and presents interesting results in a species sparsely represented in research. However, data need better representation. The main comments are below:
Introduction: somehow, the introduction does not sufficiently present the need for the study, as well as the aim and the background of the study. The first and second paragraph of the Discussion section should be maybe moved to the Introduction, or at least the authors should explain why transdermal route would be beneficial for donkeys, considering disadvantages of po and iv routes.
Figures: All legends of figures should contain the information on representation of the result: are the results presented as mean ± SD? Also, significant results in the figures should be indicated with *. Moreover, statistical test should also be mentioned.
Safety: in the conclusion, the authors mention that the drug is safe and in line 75-76, the authors mention that they observed animals for adverse effects, but these data were not presented. Please add these data.
Table 1: this looks messy, please change. Mean ± SD should be merged to one cell, as well as min and max; the column with number of animals should be deleted, as it is same for all rows. Also, column “variable” and “unit” should be deleted, and each parameter for different PK parameters should be presented under horizontal parts of table as: Dose (mg); Cmax (ng/ml); tmax (h); t1/2 (h). Here also the authors should mention statistical test and indicate significant changes between groups with *.
Figure 6B: correct the title of the figure, explain the abbreviation of 95% CI (this also for figure 5B)
Line 72: please change the term “observed”
Line 128: equation should be written in a mathematically correct way
Line 180: abbreviation BF should be used here
Article is well written and should be accepted.
Author Response
Dear esteemed editor and reviewers,
We extend our gratitude for your invaluable comments and suggestions on our manuscript. Below, we present our comprehensive responses addressing each of the reviewer comments in a line-by-line format:
Response to Reviewer 1
Comments and Suggestions for Authors
The article is well written and presents interesting results in a species sparsely represented in research. However, data need better representation. The main comments are below:
Introduction: somehow, the introduction does not sufficiently present the need for the study, as well as the aim and the background of the study. The first and second paragraph of the Discussion section should be maybe moved to the Introduction, or at least the authors should explain why transdermal route would be beneficial for donkeys, considering disadvantages of po and iv routes.
AU: We appreciate the reviewer's valuable feedback regarding the introduction and the need for the study. We revised the introduction to provide a clearer context for the study and highlight the advantages of the transdermal route in donkeys. We have moved up the second paragraph pf the discussion to the introduction.
Figures: All legends of figures should contain the information on representation of the result: are the results presented as mean ± SD? Also, significant results in the figures should be indicated with *. Moreover, statistical tests should also be mentioned.
AU: Thank you for your feedback on the figures. We have updated the legends of our figures to indicate that the results are presented as mean ± SE. We have also added asterisks to the figures to indicate statistically significant results.
Safety: in conclusion, the authors mention that the drug is safe and in line 75-76, the authors mention that they observed animals for adverse effects, but these data were not presented. Please add these data.
AU: Thank you for your feedback regarding the safety of flunixin in donkeys. To replicate field conditions, we administered the TD FM directly on the hair coat over the topline of the donkeys without shaving. We observed no adverse dermal effects, such as hair loss or skin redness, in the donkeys treated with topical medication. However, it is important to conduct additional tolerance and toxicological studies before recommending appropriate dosage rates. We have added this information in the discussion and revised the conclusion to reflect our observed data.
Table 1: this looks messy, please change. Mean ± SD should be merged to one cell, as well as min and max; the column with number of animals should be deleted, as it is same for all rows. Also, column “variable” and “unit” should be deleted, and each parameter for different PK parameters should be presented under horizontal parts of table as: Dose (mg); Cmax (ng/ml); tmax (h); t1/2 (h). Here also the authors should mention statistical test and indicate significant changes between groups with *.
AU: Thank you for your feedback on Table 1. We appreciate your suggestions for improving the clarity and organization of the table. Based on your recommendations, we have made the following changes:
We have merged the mean and standard deviation values into a single cell for each parameter.
We have replaced min and max with 95% Confidence interval.
Since the number of animals is the same for all rows, we agree that it is unnecessary to include this information in the table.
We have reorganized the table to present the parameters for different PK parameters in a horizontal format. The parameters now include Dose (mg), Cmax (ng/ml), tmax (h), and t1/2 (h), as you specified.
We have added P-values and a footnote to the table to mention the statistical test used for analysis. Additionally, we have marked the significant changes between groups with superscripts, as suggested.
These modifications will significantly enhance the readability and presentation of the table. We appreciate your valuable input in improving the quality of our manuscript.
Please see revised table 1 below in this file.
Figure 6B: correct the title of the figure, explain the abbreviation of 95% CI (this also for figure 5B)
Line 72: please change the term “observed”
AU: Changed to “allowed” as recommended.
Line 128: equation should be written in a mathematically correct way
AU: We have carefully reviewed the equation and made the necessary revisions to ensure that it is mathematically correct.
Line 180: abbreviation BF should be used here
AU: The abbreviation used as here and throughout the manuscript.
Article is well written and should be accepted.
AU: Thank you for your positive feedback on our manuscript. We are delighted to hear that you found the article well-written and worthy of acceptance. We greatly appreciate your kind words and support.
|
Route of flunixin administration |
Values1 |
AUC/Dose |
Cmax |
tmax |
Drug Half-life (t1/2) |
AUCINF |
AUClast |
|
Units |
mg |
ng/mL |
h |
1/h |
h*ng/mL |
h*ng/mL |
|
|
IV |
Mean ± SE |
1.23 ± 0.02c |
9839.42 ± 452.12a |
0.08 ± 0.00c |
0.13 ± 0.004a |
10641.68 ± 1096.733a |
10632.23 ± 1094.69a |
|
|
Median |
1.2 |
9712.673 |
0.08 |
0.137 |
9845.625 |
9838.74 |
|
95% Conf. Interval |
1.18 - 1.27 |
8885.51 - 10793.33 |
. |
0.126 - 0.145 |
8327.778 - 12955.58 |
8322.62 - 12941.83 |
|
|
PO |
Mean ± SE |
2.75 ± 0.05b |
1976.8 ± 463.63b |
1.33 ± 0.35b |
0.13 ± 0.013a |
12589.17 ± 3145.094a |
12583.5 -3144.97a |
|
|
Median |
2.7 |
2146.45 |
1 |
0.136 |
11817.5 |
11812.5 |
|
95% Conf. Interval |
2.63 - 2.86 |
998.61 - 2954.98 |
0.59 - 2.07 |
0.106 - 0.164 |
5953.599 - 19224.73 |
5948.18 - 19218.82 |
|
|
TD |
Mean ± SE |
4.38 ± 0.01a |
161.46 ± 33.89c |
5.83 ± 0.16a |
0.04 ± 0.002b |
4521 ± 787.34b |
4383.66 - 757.05b |
|
|
Median |
4.4 |
171.30 |
6 |
0.039 |
5194 |
5043 |
|
95% Conf. Interval |
4.34 - 4.42 |
89.96 - 232.97 |
5.48 - 6.18 |
0.036 - 0.045 |
2859.843 - 6182.157 |
2786.42 - 5980.91 |
|
|
P-value |
<0.001 |
<0.001 |
0.001 |
<0.001 |
0.014 |
0.013 |
1AUC/Dose: the area under the plasma drug concentration-time curve (AUC) reflects the actual body exposure to drug after administration of a dose of the drug and is expressed in mg*h/L; Cmax: is the maximum concentration of a drug in the blood after a dose is given; t1/2: is the length of time required for the concentration of a particular substance (typically a drug) to decrease to half of its starting dose in the body; AUCinf: area under the plasma concentration-time curve from time 0 to infinity. AUC was determined from the plasma concentration-time profile using non-compartmental analysis; AUClast: AUC to the Last Measurable Concentration.
a-cMean values in the same column with different superscripts differ significantly (P<0.05)

Reviewer 2 Report
Authors conducted the pilot study to examine the Pharmacokinetics (PK) of Transdermal Flunixin Meglumine (FM) as a Pain Relief in Donkeys. The study is quite interesting and useful due to the limited options are effective in equines as an anti-inflammatory agents. Moreover, PK studies are gaining importance in clinical medicine. However, I suggest following minor changes to improve the quality of the manuscript for readers and research community.
Suggestions/Comments:
Comment1. Introduction section should be improved with the incorporation of logical data. Why FM was selected as a pain killer in the presence of other options? Clinical significance of the study should be discussed.
Comment2. Line (70-71), It would be more appropriate add short reference instead of full link.
Comment3. Line 281, Authors mentioned the statistical differences among pk parameters in Table S3. Authors must check the manuscript to find out Table S3.
Comment4. Line (359- 360), latest or last stages?? Authors can modify this sentence to improve the understanding.
Comment5. Line (397), it would be better to add any reference of similar study conducted. It is suggested for authors to add more references.
Comment6. Why authors select only one transdermal dose (4.4mg)?? It would be more appropriate if authors select 2-3 doses for this route to improve its clinical significance & relevance.
Comment7. Table 1 should contain superscripts to reflect the statistical significance of the results.
Comment8. Authors should add abbreviation list in the manuscript.
Comment9. Limitation of the study should be discussed in limitation section of the manuscript.
Comment10: Authors should describe the Future perspective of the study as well.
Some minor revision required
Author Response
Dear esteemed editor and reviewers,
We extend our gratitude for your invaluable comments and suggestions on our manuscript. Below, we present our comprehensive responses addressing each of the reviewer comments in a line-by-line format:
Response to Reviewer 2
Comments and Suggestions for Authors
Authors conducted the pilot study to examine the Pharmacokinetics (PK) of Transdermal Flunixin Meglumine (FM) as a Pain Relief in Donkeys. The study is quite interesting and useful due to the limited options are effective in equines as anti-inflammatory agents. Moreover, PK studies are gaining importance in clinical medicine. However, I suggest the following minor changes to improve the quality of the manuscript for readers and the research community.
AU: We appreciate the reviewer's comments/suggestions to improve the quality of our manuscript.
Suggestions/Comments:
Comment1. The introduction section should be improved with the incorporation of logical data. Why FM was selected as a pain killer in the presence of other options? Clinical significance of the study should be discussed.
AU: Thank you for your valuable comments on the introduction. We have revised the introduction section to include relevant and logical data supporting the selection of FM as a painkiller over other options. We have provided information on the mechanism of action of FM, its efficacy in previous studies, and its potential advantages compared to alternative pain management approaches.
Please see lines 46-71.
Comment2. Line (70-71), It would be more appropriate add short reference instead of full link.
AU: We appreciate your suggestion to include a short reference instead of a full link. We agree that using a short reference will enhance the readability of our manuscript. Reference was added.
Comment3. Line 281, Authors mentioned the statistical differences among pk parameters in Table S3. Authors must check the manuscript to find Table S3.
AU: Table S3 refers to supplementary table number 3, which is provided in the supplementary materials accompanying the manuscript. If the reviewer wishes to access this table, please locate the "manuscript-supplementary.docx" file on the journal reviewer platform. This file contains the complete set of supplementary materials, including Table S3.
Comment4. Line (359- 360), latest or last stages?? Authors can modify this sentence to improve the understanding.
AU: Modified as suggested for clarity. Thank you!
Comment5. Line (397), it would be better to add any reference of similar study conducted. It is suggested for authors to add more references.
AU: We appreciate your suggestion to add a reference to a similar study conducted in the same context. We agree that incorporating additional references will strengthen the support for our research and provide a broader perspective on the topic. Here is the added similar study:
“In line with our findings, Cheng et al. (1994) observed a significant inhibition of serum TXB2 generation from 1 hour to 12 hours following IV administration of FM at a dose of 1.1 mg/kg in donkeys. However, they noted that the TXB2 levels returned to normal after 24 hours.”
Comment6. Why authors select only one transdermal dose (4.4mg)?? It would be more appropriate if authors select 2-3 doses for this route to improve its clinical significance & relevance.
AU: Thank you for your feedback regarding the selection of a single transdermal dose (4.4mg) in our study. We appreciate your suggestion for future research, and we will certainly consider it. However, it's important to note that our main objective was to compare the transdermal administration at the dose commonly used in cattle with the other routes of administration (IV and oral). This approach allowed us to evaluate the relative efficacy and pharmacokinetic profile of the transdermal route in comparison to established routes.
Comment7. Table 1 should contain superscripts to reflect the statistical significance of the results.
AU: We have revised Table 1 to include superscripts indicating the statistical significance of the results. Please see the revised Table 1 below in this document. Thanks.
Comment8. Authors should add an abbreviation list in the manuscript.
AU: Thank you for your suggestion to include an abbreviation list in our manuscript. We agree that this addition would enhance the clarity and understanding of the abbreviations used throughout the document. However, we are uncertain about whether the guidelines of the journal permit the inclusion of an abbreviation list. We will consult with the editor regarding the possibility of including an abbreviation list in the manuscript.
Comment9. Limitation of the study should be discussed in the limitation section of the manuscript.
AU: We agree with the reviewer, and we have included "Limitations section” in the manuscript to provide the readers with an overview of the study's constraints. Please see below added limitations section:
“Regarding the safety of TD FM for donkeys, we did not measure outcomes to assess the toxicity of FM in donkeys. Considering that TD FM is systemically absorbed, it implies that donkeys may be susceptible to the common side effects associated with NSAIDs administered through other routes, including gastric ulceration, dorsal colitis, and disruption of the intestinal microbiome. Therefore, further studies are required to provide crucial insights into the safety and efficacy of TD FM in donkeys, enabling better understanding and appropriate management of potential adverse effects. In our study, we only tested one transdermal dose of FM (4.4 mg/kg). However, it will be beneficial for future studies to test different doses of TD FM to understand more about its clinical significance & relevance in donkeys.
Comment10: Authors should describe the Future perspective of the study as well.
AU: Thank you for your comment regarding the future perspective of our study. We agree that discussing the potential future directions and implications of our research would provide valuable insights to readers. We highlighted areas such as exploring different transdermal dosages, conducting long-term follow-up studies, and evaluating the safety and efficacy of TD FM in larger populations.

Reviewer 3 Report
Interesting paper. Below are some suggestions to improve the current version.
The introduction is too short and need to be improved.
Here are some ideas:
I think that lines 38- 39 should be moved downward, and instead, a few line are to be added about the general approaches for pain mamnegemt in donkeys.
Lines 40-41, please add the species in which flunixin meglumine is approved.
A few lines about the transdermal system of the drug should be given upon its first mention in the paper. What is the technology? The delivery system? And what are the advantages of such route of administration in general and of this system in particular.
Also, may be a few more line about the previous results obtained with this system.
Line 115 , correct the spelling of analysis
Table one is a bit confusing, maybe it will be helpful if the borders between different variables are widened to separate them
It look strange to me the effect (onset and duration) of the drug on PGF2 was similar for the three routes. Do the authors have any possible explanation? This should be elaborated in the discussion part.
English language is fine, maybe a minor spelling mistakes are to be corrected.
Author Response
Reviewer 3
Comments and Suggestions for Authors
Interesting paper. Below are some suggestions to improve the current version.
The introduction is too short and need to be improved.
AU: Thank you for your feedback regarding the introduction. We have included additional background information, and relevant references to provide a more thorough understanding of the topic.
Here are some ideas:
AU: Thank you so much for providing this very helpful ideas to improve our manuscript.
I think that lines 38- 39 should be moved downward, and instead, a few line are to be added about the general approaches for pain management in donkeys.
AU: We appreciate your suggestions. We moved lines 38-39 down and added a new paragraph regarding the general approaches for pain management in donkeys.
Lines 40-41, please add the species in which flunixin meglumine is approved.
AU: Added as suggested. Thanks
A few lines about the transdermal system of the drug should be given upon its first mention in the paper. What is the technology? The delivery system? And what are the advantages of such route of administration in general and of this system in particular.
AU: Thank you for your valuable suggestion. We added a few lines to the introduction to explain why transdermal route would be beneficial for donkeys, considering disadvantages of po and iv routes.
Also, may be a few more line about the previous results obtained with this system.
AU: Previous results of TD FM from different animal species were added to the introduction.
Line 115 , correct the spelling of analysis
AU: Corrected. Thank you.
Table one is a bit confusing, maybe it will be helpful if the borders between different variables are widened to separate them
AU: We have reorganized the table to present the parameters for different PK parameters in a horizontal format. The parameters now include Dose (mg), Cmax (ng/ml), tmax (h), and t1/2 (h), as you specified.
We have added P-values and a footnote to the table to mention the statistical test used for analysis. Additionally, we have marked the significant changes between groups with superscripts, as suggested.
It look strange to me the effect (onset and duration) of the drug on PGF2 was similar for the three routes. Do the authors have any possible explanation? This should be elaborated in the discussion part.
AU: Thank you so much for getting our attention to this data. We found that PGF2 data was not normally distributed and needed transformation. We have updated our PGF2 alpha results and figure.